# Diverse policy maker perspectives on the mental health of pregnant and parenting adolescent girls in Kenya: Considerations for comprehensive, adolescent-centered policies and programs

Georgina Obonyo[1], Vincent Nyongesa[2], Malia Duffy[3,4], Joseph Kathono[2,5], Darius Nyamai[5], Shillah Mwaniga[5,6], Obadia Yator[2], Marcy Levy[7], Joanna Lai[7], Manasi Kumar[2] *

1 Our Voices Initiative, Nairobi, Kenya, 2 Department of Psychiatry, University of Nairobi, Nairobi, Kenya, 3 Health Across Humanity, LLC, Boston, Massachusetts, United States of America, 4 Saint Ambrose University, Davenport, Iowa, United States of America, 5 Nairobi Metropolitan Services, Nairobi, Kenya, 6 Vrije University, Amsterdam, Netherlands, 7 UNICEF Headquarters, New York, New York, United States of America

* mkumar@uonbi.ac.ke

**Data Availability Statement:** This is qualitative data with sensitive information, but is available

## Abstract

The pregnancy rate in Kenya among adolescent girls is among the highest in the world. Adolescent girls experience increased risk of anxiety and depression during pregnancy and postpartum which can result in poor health outcomes for both mother and baby, and negatively influence their life course. Mental health is often given low priority in health policy planning, particularly in Sub-Saharan Africa (SSA). There is an urgent need to address the treatment gap and provide timely mental health promotion and preventative services, there is a need to focus on the shifting demographic of SSA—the young people. To understand perspectives on policymakers on the mental health prevention and promotion needs of pregnant and parenting adolescent girls, we carried out a series of interviews as part of UNICEF funded helping pregnant and parenting adolescents thrive project in Kenya. We interviewed 13 diverse health and social policy makers in Kenya to understand their perspectives on the mental health experiences of pregnant and parenting adolescent girls and their ideas for optimizing mental health promotion. Six principal themes emerged including the mental health situation for adolescent girls, risk factors for poor mental health and barriers to accessing services for adolescent girls, health seeking behavior effect on maternal and child health outcomes, mental health promotion, protective factors for good mental health, and policy level issues. Examination of existing policies is required to determine how they can fully and effectively be implemented to support the mental health of pregnant and parenting adolescent girls.

upon reasonable request from the corresponding author and the Nairobi Ethics and Research Committee. The Institutional Research Ethics Committee of Kenyatta National Hospital and University of Nairobi prevent sharing this sensitive data as several policy makers were consulted. The point person in charge of the ethics body is Dr Beatrice Amugune who is the chair of the IRB. The IRB can be contacted for accessing data on reasonable request and the address and contact details are Kenyatta National Hospital-University of Nairobi Ethics and Research Committee P.O Box 19676-00202 Nairobi Email: uonknh_erc@uonbi.ac.ke.

**Funding:** Research reported in this publication was supported by the Fogarty International Center of the National Institutes of Health under Award Number K43TW010716, which also supported the contributions of MK to this work. The content is solely the responsibility of the authors and does not necessarily represent the official views of the National Institutes of Health. The qualitative inquiry is part of an embedded study Helping Adolescents Thrive (HAT) supported by UNICEF Headquarters. The funders had no role in study design, data collection and analysis, decision to publish, or preparation of the manuscript.

**Competing interests:** The authors have declared that no competing interests exist.

## Introduction

In Kenya, the pregnancy rate among adolescent girls ages 15–19 is among the highest in the world with 82 births/1,000 adolescent girls, which will lead to an estimated 16.4 million adolescent mothers by 2030. [1, 2] Global study findings demonstrate that between 8.3 to 39% of pregnant adolescent girls experience depression in comparison to adult women, among whom 11–18% are estimated to experience depression [3–5]. A World Health Organization, UNICEF Lancet Commission published in 2020 on the *Future of the World's Children* highlighted the clear responsibility that governments have to care for children across all sectors, underscoring that investments in the health, education, and development of children not only have positive impacts on the child, but also have positive intergenerational and societal impacts [4]. Given the evidence that poor mental health (MH) during pregnancy and postpartum leads to inferior short and long-term health outcomes for both mother and baby [5–7], it is imperative to examine the policy environment to determine how it can better support the mental health of pregnant and parenting adolescent girls.

From a policy perspective, there are multiple opportunities to disrupt pathways to poor mental health among pregnant and parenting adolescent girls by learning from well-documented social determinants. Using Consolidated Framework for Implementation research, evidence on inner and outer setting factors prevalent in the East African region demonstrates that stigma and shame from the family and community for becoming pregnant, poverty including experiences of food insecurity and inability to purchase basic needs for their children, poor treatment from health workers, and inadequate access to health facilities all significantly impact mental health [8–11]. Similarly, reduced access to education is a pervasive issue with negative mental health impacts [8, 9, 12]. In the most recent Kenya 2014 Demographic Health Survey, there were approximately 13,000 adolescent girls required to drop out of school due to pregnancy, resulting in reduced educational attainment, earning potential, and increasing risk of early marriage [13, 14]. Appraisal and review of policies that seek to address the roots of poverty for pregnant and parenting adolescent girls and its impacts including food insecurity, and access to education and healthcare may help to prevent mental stress during the perinatal and postpartum periods [8, 15, 16]. An approach that addresses determinants of adverse outcomes for pregnant and parenting adolescent girls along with a reflection on the process of policy making and mental health friendly public policy/decision science will seek to reduce the inequities encountered in this population.

Enshrined in Kenya's constitution, including within the 2007 Reproductive Health Policy, adolescent girls have the right to healthcare including access to adolescent friendly health services that are meant to be offered free of charge across all health care service contexts [17]. While there is no age of consent required for access to reproductive health services [18], access is severely limited with only 12% of health facilities offering reproductive health services to adolescent girls. These access issues are further aggravated with stigma and opposition to sexual and reproductive health rights in school context [8]. Multilevel stakeholder involvement including provider, community, teacher, and parents' attitudes, negligence, and other system level challenges, aggravate unmet sexual, reproductive, mental health and psychosocial needs of adolescent girls [19]. Policy considerations to address provider attitudes towards adolescent girls seeking reproductive health services is described as a priority for the Government of Kenya [17]. In addition, policies that support and enforce broad implementation of adolescent friendly services, including respectful treatment of pregnant and parenting adolescent girls within antenatal and postnatal services may also be beneficial [8].

This study took place within the context of World Health Organization's and UNICEF's Helping Adolescents' Thrive (HAT)- Kenya initiative and is also nested within the NIH funded

'Implementing mental health interventions for pregnant adolescents in primary care LMIC settings' (INSPIRE) study in Kenya. The focus of both studies is on peripartum adolescent girls. INSPIRE focuses on developing treatment interventions for this cohort. HAT provides strategies, guidelines, and tools to promote and protect adolescent mental health and reduce self-harm and other risk behaviors. This study examines the perspectives of diverse policy makers on mental health promotion and prevention among pregnant and parenting adolescent girls to help identify points of entry and barriers to developing policies, programs, and messaging for mental health promotion among peripartum adolescent populations. Study findings can be compared with findings documented from a separate set of key informant interviews among peripartum adolescents ages 13–24 [20], to help strengthen policies that can influence the mental wellbeing of peripartum adolescents in Kenya.

## Materials and methods

### Study design

This descriptive research draws upon a qualitative approach, to collect data via semi-structured interviews among stakeholders involved in health, social and educational policy that influences pregnant and parenting adolescent girls in Kenya.

### Recruitment

Participants were recruited from the INSPIRE study's existing technical advisory board of policy, academic and community advisors. An invitation email with details of the study purpose and aim of interview and consent form was circulated to 20 individuals. Follow up phone calls were made to those who had not responded to the invitation email. In total 13 agreed to participate (35% refusal rate). Key informant interviews were conducted with policy makers from different sectors including the Ministry of health (MOH) such as those from Division of Mental Health, the Division of Community Health, representatives from Nairobi metropolitan services, clinical services in-charge at the national teaching and referral hospital and medical superintendent of Nairobi County maternity hospital. We also interviewed two regional representatives from UNICEF regional office who provide technical assistance to the ministry of health for programs targeting pregnant and parenting adolescent girls. Among the 13 key informant interviews, the point of saturation was reached evidenced by similar information emerging across the interviews which were conducted at convenient timings of our respondents. Consent to record the conversation was sought from the participants and the recordings were saved on a password protected computer and transcribed verbatim by a research team member.

### Settings

Due to the ongoing remote working and social distancing procedures due to COVID-19 in the Government of Kenya offices we decided to carry out online interviews. Prior to the interviews, a study team member, VN, contacted the interviewees and took them through the remit of the study, briefly described the interview guide, and the process of consenting. Both oral and written consent was taken, the consent forms were scanned and emailed to respondents in advance.

### Data collection

Following consent, key informant interviews (Table 1) were conducted in English using an interview guide and recorded over Zoom. The interviews were led by a youth advocate, GO, along with researchers VN, JK and MK. Googlejam boards were populated during each interview by VN, MK; the interviews were conducted by VN or GO with participation from JK,

**Table 1. Interview guide for policy makers.**

*Interview guide for policy makers on adolescent health and mental health specialists*

**Barriers-**

 1. How do we address mental health needs of pregnant adolescents?

 2. What do you think are commonly experienced barriers they encounter?

 3. How do mental health related barriers impede care seeking behaviors?

 4. How do those impact infant outcomes?

 5. Is policy development/lack of a barrier?

 6. Do you think all programmatic and policy structures are in place to support an integrated ANC-PNC inclusive of mental health?

**Solutions-**

 1. Who are people that can help in bringing the change?

 2. Do you know of solutions or exemplars that have worked in your programming/department?

 3. What solutions you would like to implement?

 4. Where does mental health figure in there?

 5. What role does drug and substance use prevention play there?

 6. Can simple socioemotional learning tools help?

 7. What role do you see of problem solving and peer support in providing timely and effective mental health care during ANC/PNC?

MK. The Jamboards were visible to the interviewees so that they could think through their responses and give real time feedback on whether their information was well-captured. A sample Jamboard is given in Fig 1.

## Data analysis

The interviews were uploaded on the Qualitative Solution for Research (QSR) NVivo version 10 software. Adequate familiarization with the data was done by thorough reading through all the transcripts prior to the coding process by the research team. Thereafter, taxonomic data

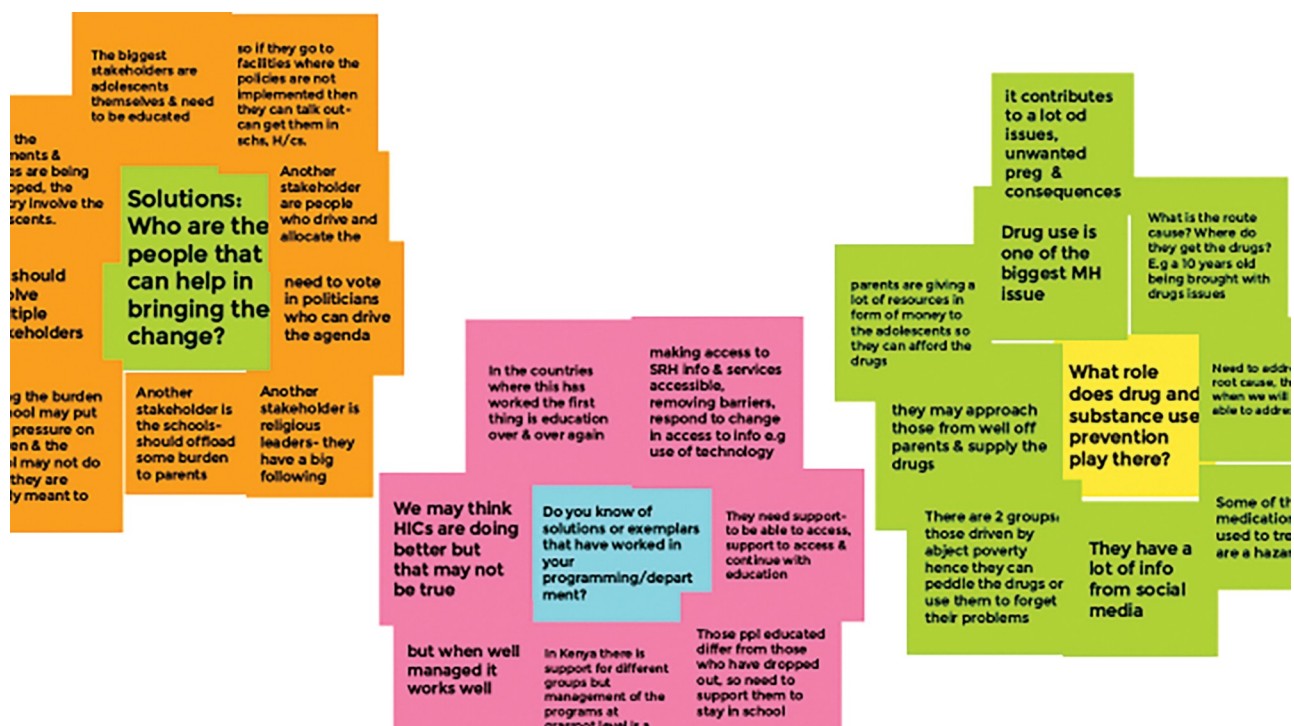

**Fig 1. Sample Jamboard.**

classification was conducted through line-by-line coding. Thereafter, a general coding framework was developed for all the categories. The categories were further analyzed, and the emerging dominant overarching patterns of respondents' experiences and viewpoints were identified as key themes while the lower categorized hierarchy formed the sub-themes. Cross tabulation, word frequencies and coding matrix queries were conducted during the analysis. The analysis was an integration of inductive and deductive approach. The key researchers, MK, VN, GO, JK discussed the interviews and the themes/sub-themes to further review coherence of findings against the overall experience of the interviews. Comparison of key themes identified across the separately published adolescent study and this policy study to identify common priorities and areas where there were differences in perspectives.

### Theoretical framework

This study methodology revealed a dearth of models applicable to low-income countries that provide a framework to examine mental health policies. To address this, we applied an integrated framework, bringing together the Consolidated Framework for Implementation Research (CFIR) [21, 22] with the Analysis of Determinants and Policy Impacts (ADEPT) model [23] and the Knowledge, Policy and Power Framework (KPP) [24] The CFIR framework was used to help elucidate the societal and organizational factors that influence mental health policy implementation. Determinants from the ADEPT model were integrated into the CFIR framework in addition to an examination of the political context as revealed through interviews as described within the KPP (Fig 2). The results were synthesized within this integrated framework to provide an in-depth understanding of the mental health policy environment for pregnant and parenting adolescent girls within the study context.

### Ethical clearance

The study was approved by Kenyatta National Hospital/University of Nairobi ethical review committee (Approval no. P694/09/2018). Approval was received from Nairobi County Health no. CMO/NRB/OPR/VOL1/2019/04 and a research permit from the Kenyan National Commission for Science, Technology and Innovation (NACOSTI/P/19/77705/28063) was obtained. All participants gave consent for the information to be published.

### Results

Thirteen policymaker key informants participated in this research (for the purpose of confidentiality we decided against providing too many details about them). Six principal topics emerged across the key informant interviews including the mental health situation for

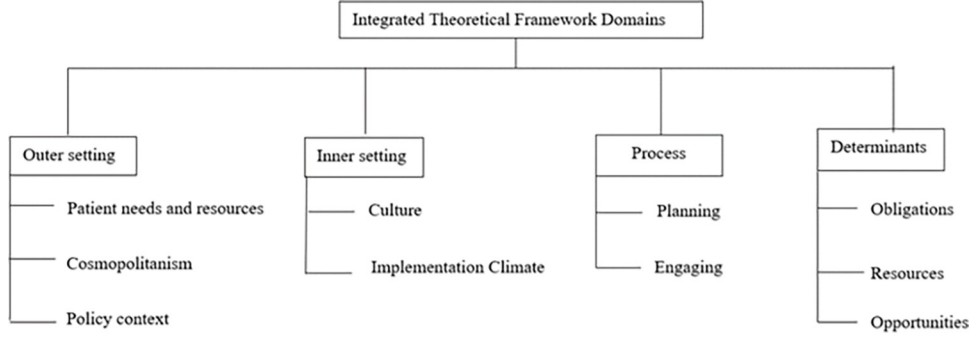

**Fig 2. Integrated theoretical framework.**

adolescent girls, risk factors for adolescent poor mental health and barriers to accessing services, health seeking behavior effect on maternal and child health outcomes, mental health promotion, protective factors for good mental health, and policy level issues.

## Mental health situation for adolescent girls

Informants identified depression, suicide, and substance use disorder as the most common mental health challenges. Depression was identified as a common barrier to accessing antenatal care (ANC) services.

*"We have experienced an escalation of suicides among this population in the community, and we have seen it and we are not adequately addressing it, and therefore a depressed girl will not actually be proactive to go and look for those services, they don't even know where to look for mental health services." (P5, female)*

Substance use was identified as a contributor to unwanted pregnancies

*"Drug or substance use is one of the big mental health issues we are dealing with currently, and I think it is the environment, and it is contributing to a lot of the issues and a lot of the unwanted pregnancies and the consequences of that." (P5, female)*

Informants also recognized that the pregnancy itself is a source of mental stress and accompanied by the fear of likely having contracted a sexually transmitted infection such as Human Immunodeficiency Virus (HIV).

*"There is also that feeling of loss for an adolescent loss of freedom, loss of you know, that being young now you are going to be called a mother, so that also affects them and a lot of them get to get psychological issues and you know they even get into depression, and some get even to a suicidal ideation." (P2, female)*

## Risk factors for adolescent girls' poor mental health and barriers to accessing services

Informants described adolescent pregnancy as being perceived as a deviation from cultural and societal norms.

*"The breakdown of the parental or guardian relationship as a result of feeling of disappointment due to the, you know culturally, the expectation is that people are married, and they are older before they get children." (P2, female)*

Adolescent pregnancy and single motherhood are perceived as failures to live up to society's expectations, potentially resulting in fractured friendships and broken ties with religious organizations, resulting in stress.

*"For those who are religious, also there is that rejection from friends and religious affiliations, because of the feeling that it is not acceptable." (P2, female)*

*"These are young adolescent pregnant, and then eventually becoming mothers they will have a lot of stigma within their society, because of our sociocultural setting, family expectation, community expectations." (P12, male)*

It was also widely acknowledged that adolescent girls face challenges accessing information and that they prefer to access information digitally.

*"Therefore, there is one aspect of power play, and therefore access even to just information about their sexual and reproductive health rights and their mental health rights, a lot of the young girls do not access this both in the community and in schools." (P5, female)*

*"We know that this is also an era of technology, and a lot of these people would want to have that information as e-Information, but there is also the barrier of access, and also we have not yet developed a lot of these materials to deliver it in a youth friendly manner to them." (P5, female)*

Stigma for becoming pregnant and the ensuing social isolation were identified as significant contributors to mental stress. Informants noted that there were often different levels of stigmatization for becoming pregnant including from oneself, family, the community and from health care workers, which is compounded when adolescent girls experienced sexual violence.

*"Socially they are already feeling guilty as I said, so they feel unaccepted, some which actually maybe the truth, they have already been rejected at their homes so they are trying to see where they fit in, and then they come to a hospital where they feel again people are not understanding them, so socially there is that feeling of they don't belong and they have done wrong and they would rather keep off." (P9, female)*

Negative attitudes from health care workers were thought to repel adolescent mothers from seeking services.

*"So if the services are not acceptable, they're not youth friendly services, especially where they are getting them, then they will not go for them because they will expect probably the nurses, the health workers there will not treat them well, or they will be together with the adult population as they seek these services, so the services then will not be acceptable, and they will not go for it." (P12, male)*

Clinicians' lack of competence to provide mental health care to adolescent girls was noted, including mental health screening and communication. Negative attitudes of health workers, lack of youth centered services, poor evidence informed programs were alluded to.

*"One of the barriers for access I would say is skills and competencies among our health care workers in dealing with pregnant women, or pregnant girls- adolescent pregnant girls in terms of the health care workers realizing that this is already a special population, and the specific needs that they have more so when they are pregnant, and they susceptibility to mental or perinatal mental health illnesses. So one of them is lack of that skill." (P4, female)*

Informants also expressed that clinicians are not adequately equipped to deal with their own stressors which negatively affects how they provide care to their clients. There is also a lack of competencies for clinicians to link with the criminal justice system in instances where adolescent girls experience sexual violence.

*"We have also noted that we need a lot of support in the area of mental health for the health care workers themselves. I think once I say a saying saying Physicians heal thy self, and I think that was coined by the fact that there is a lot that is expected from us but nobody looks in to*

*think about our needs. In this time of COVID there has been a lot of mental issues among our staff, and we believe for you to offer service optimally you also need to be healthy." (P9, female)*

Informants recognized challenges associated with lack of collaborative care including access to the legal system when it is needed. Poor service integration was also identified, noting that mental health is not identified as a priority or often integrated into ANC.

*"The other barrier is that lack of integration of perinatal mental health services, or maternal mental health services in the normal primary health care and starting now for example, in this case we are talking about pregnancy so we look at ANC, it is not a routine; the mother will be checked about the blood pressure, about HIV and that is for prevention of PMTCT, blood level, the HB level, they will be given information about nutrition, how to check for the danger signs, but when it comes to maternal mental health, the danger signs, or how to identify, or how to keep mentally healthy during this period it is not integrated and so it is not given any prominence." (P4, female)*

*"A lot of the girls, the intercourse was either not consensual, or it is with a family member and therefore, even just looking for legal services and the stigmatization and intimidation, especially from our police force, and the other people who deal with those issues can be a main barrier, and also the fear that they may not continue getting the societal support, the family and the bigger societal support once they report their problem." (P5, female)*

Economic challenges were identified as a pervasive issue, which our informants underscored strongly. Poverty is a significant barrier to accessing optimal health care and meeting the cost involved in bringing up a child.

*"Economically, remember these are people who have either dropped out of schools or they are not working, they are depending on probably their parents or their boyfriends who are also not probably you know capable, so they feel quite unprepared to meet the financial burden that comes with bringing up a baby." (P9, female)*

Informants identified costs associated with health visits to include transport, laboratory charges and buying some commodities like gloves, cotton wool which might not be available in the "free" ANC facilities.

*"Of course if they are adolescents and have no income generating activity, then there is the financial barrier to access and this will come about that even though our facilities offer free antenatal and delivery services at particularly level two and three, then there are other costs associated with seeking care, and these would include issues around transportation to be able to move from one end to the other in cases where they are prescribed for supplements that are not available then they'd have to also meet that cost." (P2, female)*

Informants noted that the shortage of mental health specialists including psychiatrists, psychologists, and mental health nurses is a significant barrier to providing mental health services and the potential for training and task shifting to address shortages.

*"I think up to the last three or so months we didn't have psychologists like here in Pumwani, we didn't have a psychiatrist for the longest time, and even the one we have is a shared one, I mean she has to go to other facilities as well. So in terms of human resource capacity, I feel we*

*have not been- it is something that has really not been given a lot of emphasis yet we need it."
(P9, female)*

*"Having skilled personnel as a huge gap in our country, the skilled mental health professionals are not commensurate to the population, we have very few skilled mental health experts." (P7, female)*

Gaps in the data management system include a lack of stratified data and poor reporting for age-disaggregated ANC and mental health indicators, particularly for suicide, keeping youth in mind.

*"About the adverse birth outcomes in adolescent mothers, to be honest when we are keeping track like we are not able currently in our reporting system at national level, because our reporting system is the Kenya health information system, so yes it will track the pregnant mothers attending adolescent clinic, the proportion of adolescent mothers attending antenatal clinic, but we are not able to specifically tell this number of deliveries were attributable to adolescent mothers and more so this number of neonatal deaths stratified according to causes are not grouped according to the age of the mother." (P7, female)*

*"The issue of data, and how best we can use data to inform programming, it would be great if all stakeholders think through how data can be meaningfully collected, to inform us of what is going on. Why I bring this up is I mean, very recently there was this huge report of the number of suicides that are going on in our country, and I was just thinking to myself, how well was this known to the health system? Because it came from an administrative report; the police report." (P11, female)*

## Health seeking behaviour effects on maternal and child health outcomes

Informants noted that adverse maternal and child outcomes result from delays in seeking healthcare and lack of access to perinatal care.

*"If the mother's mental health needs are not met, they may get into severe depression or other severe mental health conditions, and obvious when their mental health is more affected and they end up giving- when they give birth we know the risk of even getting into more severe complication in terms of their mental health; postnatal depression and psychoses, the way they have the developing of the bonding and attachment with the child which are key elements for the child growth and development." (P12, male)*

Maternal related adverse effects included postnatal depression and psychosis, preeclampsia, anaemia, and premature labour. Child related adverse effects include neglected babies, cerebral palsy, retarded growth, increase mortality, low birth weight, mother to child transmission of HIV, birth defects such as spina bifida, maternal depression affects breastfeeding, and infanticide.

*"During pregnancy in the you know, antenatal period, if we are already dealing with stigma-related to mental health, then we are unlikely to see this person even appear for normal and focused antenatal care, now that will obviously, then we cannot identify any issues that need to be addressed as early as possible. We will also see you are likely to also have nutrition-related complications, and if she is not eating well, if she is not sleeping well then it of course affects the infant's- the fetal growth and this can lead to premature labor, it can lead to anemia and other factors that would cause other complications during delivery." (P11, female)*

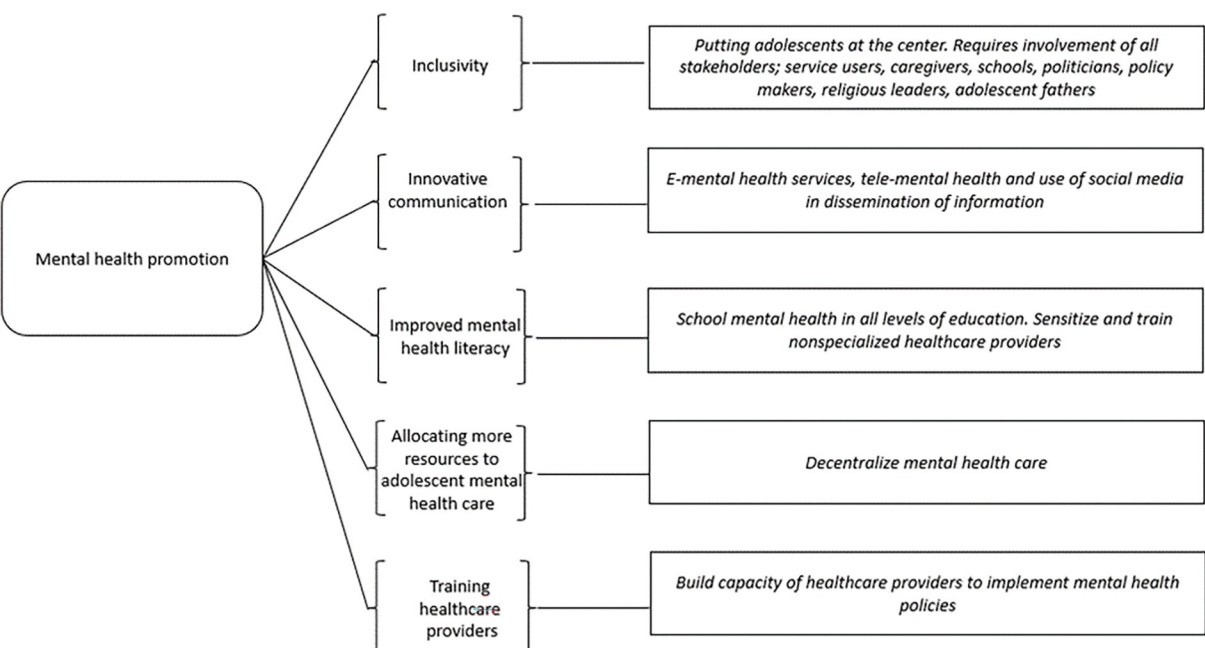

**Fig 3. Key elements of mental health promotion identified by our informants.**

## Mental health promotion

Mental health promotion opportunities identified by policy key informants (Fig 3) focused on five elements to strengthen mental health promotion opportunities among pregnant and parenting adolescent girls.

The importance of inclusivity was emphasized in the development of key policy documents, noting that inclusion of adolescent girls is paramount.

*"I tend to think at the point of developing these policies it is good to involve the relevant stakeholders and I believe the greatest stakeholder in this case is the youth themselves because they know what it is that affect them and then they can resonate with their needs better than if it is an adult who is just sitting somewhere and coming up with ideas on what should be done as far as adolescent and youth health is concerned." (P9, female)*

Additionally, inclusion of religious leaders, policy makers, youth mental health advocates, mental health care service providers, educationist, Ministry of Labour and Social Services, Education and groups representing caregivers, community health workers, politicians, adolescent fathers was also recognized as important.

*"All stakeholders and starting from the community going on to the service provisions team, you know, other stakeholders, CSOs, you know, advocates for young people, and mental health, and even policymakers. So, it cuts across it is everybody who has to come together." (P2, female)*

Innovative communication was recognized as important to reaching adolescent girls with mental health services and messages including through utilization of e-mental health services, tele-mental health, social media, and media personalities.

*"You can use tele-mental health services where you are using the technology use of telephone other internet services, in which way you can connect very well with these teenagers or adolescent pregnant mothers. You know most of them may have access to mobile phones or any other internet services'. …….. through e-Mental health services and give them the confidence even to be able to have a one to one interaction later on after they have a rapport." (P12, male)*

Improving mental health literacy was recognized as important including through integrating mental health literacy in schools from primary level to tertiary level, sensitizing and training non-specialized health care providers. Informants also noted that the aim of mental health literacy efforts should focus on addressing stigma and consequently improving access to mental health care.

*"We can address this (stigma barrier) by trying to generally bring awareness about issues of adolescent pregnancy, how much impact and trauma they may have depending on how we react to that experience, to their families, to their mothers, to their dads, you know, having sessions with those family sessions, the school sessions, so that can address that issue of stigma and the consequences it comes with." (P12, male)*

*"I think really investing in communities to understand mental health issues and needs and support and that it does not need to be discriminated against or stigmatized at community level is critical." (P11, female)*

Informants suggested allocating more resources to adolescent mental health care through decentralizing mental health care and ensuring that counsellors and psychologists are available at primary health care (PHC) facilities. Training Mother and child health (MCH) teams on integrated packages for adolescent mental health was also recognized as an approach to increase availability of mental health care for adolescents.

*"Capacity-build or rather to disseminate on the same to see how to have integrated models for mental health support for adolescent and teenage pregnancies, and also then to capacity-build more personnel to be able to provide these services and to focus more, instead of having stand-alone psychologists, but to focus more in ensuring the MCH teams then are trained on some of these interventions." (P2, female)*

*"Training the people within the communities and also integrating mental health services within where they are getting maternal and child health services would be very key, so that those people not only provide maternal and child health or any other reproductive services, but they are also able to give mental health services." (P12, male)*

*"Efforts put towards training on maternal mental health, because the ones that I have done you realize a nurse who is very good in delivery has no orientation in terms of maternal mental health, so we need to have specific- the trainings tailored we are talking about maternal mental health, so that the nurse in Pumwani when a mother walks in they have that lens of maternal mental health." (P4, female)*

### Protective factors for good mental health

Support to continue education for pregnant and parenting adolescents was widely regarded as critical to protect mental health with informants noting that getting adolescent mothers back into school is beneficial for their mental health.

*"Access and support to stay in school is one of the big determinants of health for these young girls." (P5, female)*

*"There is already a return to school policy for pregnant adolescents where they are encouraged immediately they give birth they go back to school." (P3, female)*

Integrated youth friendly programs were also widely recognized as having potential to be beneficial for adolescent mental health. Informants noted that characteristics of integrated youth friendly programs should focus on adolescent girl's needs and provide all services under one roof including utilization of mental health screening tools, and a care package to address harmful substance use while making maternal mental health programs more visible among other services.

*"Try to build youth friendly services, and this one, you know, our services are more set up for adults. So setting up services is not an easy thing, it is something requiring a lot of investment. So we have to take advantage of already existing services and try to see how can we incorporate some youth friendly kind of concept." (P12, male)*

*"You shift the practice from the office practice where you sit and people find you to find people where they are." (P12, male)*

Implementation of peer support was also identified as important to facilitate sharing of experiences which can be helpful for adolescent girls to normalize what they are experiencing as a stressful situation.

*"Adolescents they are at this stage where they look to their peers for information, so the younger child would be looking to you know of course their parents, and then as they grow older around six to ten years old they are looking at the teachers, but in the age of adolescence adolescents actually consider information from their peers more superior than, or more accessible than whatever their parents or their teachers are telling them." (P3, female)*

Addressing alcohol and substance use disorders was identified as critical as informants noted, alcohol and substance use affects health seeking behaviors. Informants also noted that timely mental health intervention is protective of adolescents seeking out harmful alcohol and substance use as coping mechanisms.

*"There's a coping mechanism; the young people also if we don't address their mental health challenges, they may resort to using substance to cope with the stresses they are going through. So that is another area that needs to be addressed. So the more you make a mental health friendly society, the rest you are able to address the issue of substance." (P12, male)*

Developing an impetus towards basic problem-solving skills among adolescent girls to deal with adversities was noted to help them develop their own solutions and address psychological stressors.

*"What I can say is that we have a model; PM plus, problem management plus, which is a WHO model which has been piloted and tested and it is working, people have been supported; it is a psychological model that help adolescent girls- adolescent male and girls to restructure themselves in life again and come up to look at their own socioeconomic life, to make sure that they make their own decisions in a more positive manner." (P10, male)*

Facilitating socio-emotional learning was also noted to help adolescent girls deal with stress, managing emotions and making informed decisions including to avoid suicidal behavior.

*"Yes, it (socioemotional learning tools) has been demonstrated to be very useful in various interventions; suicide prevention, in issues of substance use prevention, and many others. So yes, that is element we need to have within especially now in our school environment." (P12, male)*

Building up the family as a source of support for adolescent girls was noted to potentially be a mechanism to instill confidence and promote health seeking behaviors.

*"Empower the parents so that they are able from a very early age, to engage their children and to walk through with them from childhood through the adolescence period in confidence." (P5, female)*

Strengthened community mental health programs and advocacy were also identified as important approaches.

*"Other institutions we give guidance that they should routinely offer you know, counseling services and things like that for those who need, at the community level there are these community mental health programs." (P3, female)*

*"Other people like the teachers, like I say, religious leaders, anyone who has some sphere of authority, or that can offer help at community level, should be well sensitized to offer information or access to mental health." (P11, female)*

## Policy level issues

Informants identified aspects of the existing policy level framework that support pregnant and parenting adolescent girls including the adolescent package of care (adolescent health policies), child and adolescent health policy, childhood development policies, mental health policy, integrated early childhood development policy, the return to school policy for pregnant adolescents, and drug and substance use guidelines.

*"If you look at the policy guidance from the Ministry of Health, then there are very clear policies around antenatal care and there is even an adolescent and young people sexual reproductive health policy, and even in the county, we've been able to customize and have one for Nairobi that is specific on that." (P2, female)*

*"There are already policies addressing mental health issues in the country, there are already policies addressing antenatal care for pregnant women, and also adolescent health policies. So the issue here is that when it comes to that integration, perhaps the policy is not as clear on the same." (P2, female)*

Informants noted that the key policy challenge is the actual implementation and timely policy response. Relatedly, other barriers include lack of ownership by end-users, unclear policy especially the mental health policy and sexual and reproductive health and rights (SRHR) policy targeting pregnancies in young girls. Additional policy challenges include rushed policy framing without adequate time for implementers to understand its length, breadth, indicators for tracking progress and resource allocation. Shortage of trained personnel for

implementation, unclear costs associated with implementation (a perception that mental health programs entail high costs), and lack of streamlined health promotion programmatic approach present further barriers. Surprisingly we learnt that the biggest elephant in the room was that at times policy being a barrier in itself, if it did not appreciate contextual issues or was developed without the needs and preferences of adolescent girls at the center. It was also noted that the organigram in the MOH reduced mental health to a division and therefore it has little influence in pushing agendas and unless policy and political level empowerment of relevant departments/divisions happen the benefits would not trickle down to needy populations.

*"I can say the programs are there, the policy directions are there, but specifically one addressing mental health in that programmatic area, in that policy area this is where the gaps are, and this is what I initially said, if the experts on mental health who are mental health experts are not sitting in that on that table where adolescent health issues are being discussed, if they are not on that table so the documents which comes out from those discussions actually lack areas to address mental health." (P8, male)*

*"I think the gap is in terms of translating policy into decision making and implementations and outcomes. And here the key element is on the issues of adoption of the policies by the various sectors, whether is in education, whether it's in health, whether it's in the community, and then once they adopt those policies, do they make the necessary decisions on what they need to do and implement them?" (P12, male)*

### Theoretical framework synthesis

Deeper analysis of study findings as examined through our integrated theoretical framework identified key enablers and barriers to MH policy implementation (Table 2). Given that none of the frameworks alone capture the complexity, in combing these we can find synergistic appraisal of multilevel barriers and opportunities that can be unpacked.

## Discussion

The risk factors for poor mental health identified by key informants in this study closely aligned with the identified contributors to poor mental health from the perspectives of pregnant and parenting adolescent girls which the study team identified in the separate arm of this study, which will be published elsewhere. Most notably, economic challenges were very often noted among pregnant and parenting adolescent girls as having a significant influence on their mental health, due to inability to purchase food for themselves or their infants, purchase essential items such as diapers, or to attend doctor visits or attend school. Food insecurity has been reported elsewhere in the literature as a significant contributor to depression among pregnant and parenting women, including adolescent girls [25–27]. In this study, policy informants noted economic challenges in regards to the adolescent girls' dependence upon partners and caregivers, the costs associated with raising a baby, and costs associated with clinic attendance such as transport and laboratory fees as the most compelling financial demands for young mothers. Policies that help address food insecurity among pregnant and parenting adolescent girls as a means to support mental health [27]

Our policy key informants also identified stigma and social isolation as significant risk factors for depression, which were also commonly identified by pregnant and parenting adolescent girls who described experiences of forced eviction from their homes, premature forced departure from school, partner and friend abandonment, verbal harassment from community members, and poor treatment from healthcare workers. These findings are in alignment with

**Table 2. Integrated theoretical framework for system and policy level appraisal of adolescent mental health.**

| Outer Setting | Inner Setting | Process | Determinants |
|---|---|---|---|
| *Patient needs and resources*: Implementation of mental health policies that support pregnant and parenting adolescent girls was widely acknowledged as a critical gap. *Cosmopolitanism*: Participants acknowledged a need for further linkage between diverse institutions and stakeholders. *Policy context*: Participants noted that policies are in place including the adolescent package of care (adolescent health policies), child and adolescent health policy, childhood development policies, mental health policy, integrated early childhood development policy, the return to school policy for pregnant adolescents, and drug and substance use guidelines. The pervasive challenge is their implementation. At the systems level this includes insufficient funding for their full implementation and a lack of age-disaggregated data within ANC to fully understand the specific needs of pregnant adolescents. | *Culture*: Despite adolescent friendly policies, stigma towards pregnant and parenting adolescents is perceived to interfere with their implementation. *Implementation climate*: Participants acknowledged the need to expand policies that effectively support the mental health needs of pregnant and parenting adolescent girls. However, there are clear gaps in available funding, human resources, training, and enthusiasm to implement the policies in communities and facilities. Among pregnant and parenting adolescent girls, pervasive poverty and stigma further contribute to demand-side challenges. | *Planning*: Some policy documents include mental health considerations for pregnant and parenting adolescent girls, however, there remain further opportunities to develop cohesive, supportive policies and to develop detailed implementation plans that consider required funding, training, supportive supervision, and that ensure that mental health is integrated and decentralized at service delivery levels. *Engaging*: While there are some mental health champions, multi-level stakeholder engagement from policy makers, faith leaders, youth advocates, educationists, service providers, families, adolescent fathers and others is a critical need. | *Obligations*: No discrimination is a universal right for children; however, pregnant and parenting adolescent girls experience pervasive stigma and discrimination in this context. Governments are also obligated to ensure that children experiencing poverty receive financial support, indicating that poverty barriers to mental health support are obligated. Governments are also obligated to ensure that children have access to health care, including mental health. Stakeholders widely acknowledged poverty as a critical barrier to mental health service access. *Resources*: There are insufficient resources including funding, human resources, and capacities to implement mental health policies that support the needs of pregnant and parenting adolescent girls. *Opportunities*: The political climate is not fully motivated to broaden mental health policies that support pregnant and parenting adolescent girls. There are significant societal barriers to overcome at the policy and practice levels to create opportunities for policy development and full implementation. A groundswell of public support is also lacking, in part due to stigma associated with adolescent pregnancy. |

earlier studies among pregnant and parenting adolescent girls in Kenya which similarly identified stigma and discrimination, including social isolation as significant contributors to poor mental health [8, 9, 11, 28].

While there is dearth of literature on intersectional experiences of stigma for pregnant adolescent girls experiencing symptoms of mental illness, studies that examine intersecting factors for people living with HIV including socioeconomic status, race, gender and other factors have found compounded barriers to healthcare access, education, employment and housing [29]. Pregnant adolescent girls in Kenya report profound experiences of societal, family and self-stigma. In addition, stigma towards mental illness is pervasive, not only in Kenya, but globally. Studies in Kenya among pregnant adolescent girls have found that depression and stigma present barriers to HIV service access [30]. Given the evidence of the negative impacts of intersectional stigmas in other areas of research, it is reasonable to expect that pregnant adolescents experiencing symptoms of mental illness may experience intersecting stigmas that can contribute cumulative experiences of oppression that impede their ability to access their right to health and education. Policy makers seeking to codify youth-friendly services must consider the significant training and sensitization of health care workers required to actually create welcoming and accepting environments for pregnant adolescents experiencing depression.

Notably, pregnant and parenting adolescent girls often talked about mourning their missed life opportunities, and particularly their inability to return to school (either due to childcare needs or economic challenges) (from our experience as researchers). However, this issue was

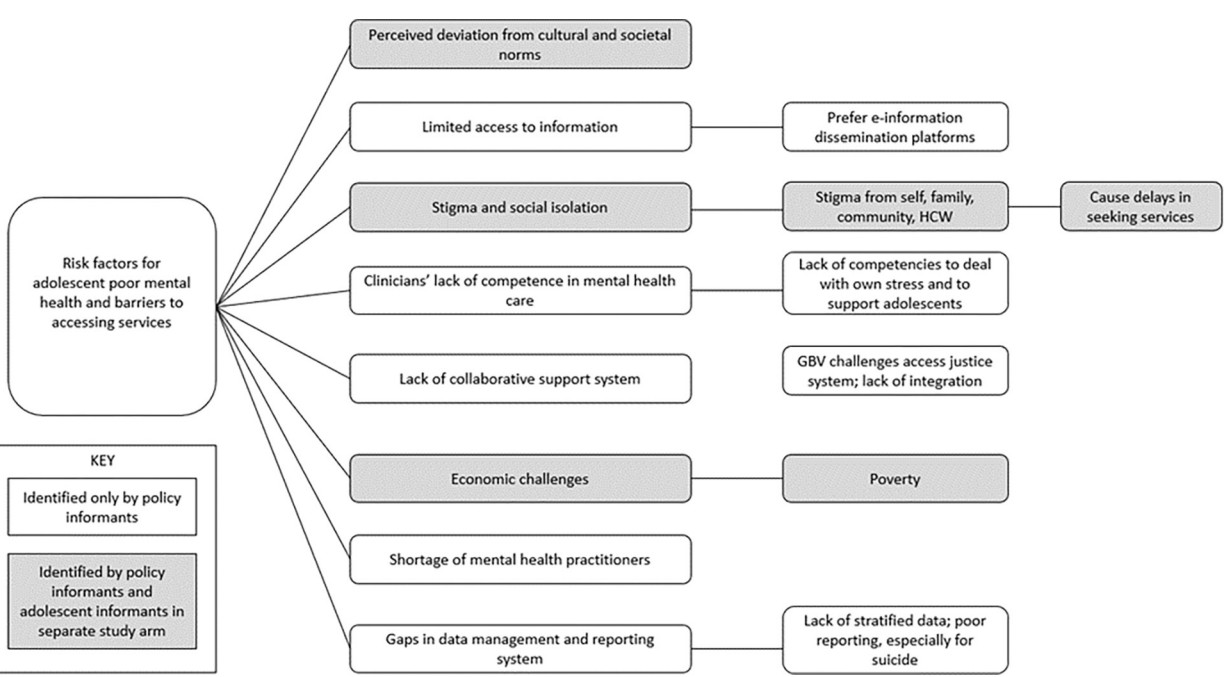

**Fig 4. Overlap in identification of risk factors and their key notes by policymakers and adolescent girls.**

not highlighted by policy informants as a contributor to poor mental health. School attendance discussions by policy makers was limited to only note that a return to school policy in place despite the fact that logistics surrounding breastfeeding and childcare are significant impediments. Other studies have also highlighted that, though parenting adolescent girls may return to school, significant barriers exist including full implementation of the policy by schools, cost, attending to the academic needs of those who have fallen behind their classmates, and poor treatment towards parenting adolescent girls within schools [16, 28, 31, 32].

The broader system wide decisions tended to dominate the thinking of our participants than micro-experiences of adolescent girls living through unplanned pregnancy and parenthood. While pregnant and parenting adolescent girls most often noted challenges that immediately influence their mental health, policy informants were more likely to identify systems issues such as shortages of mental health specialists, data management challenges, and lack of mental health competencies among healthcare workers as issues (Fig 4). This sequential inquiry in our thought process as researchers who have worked with adolescent girls to now focusing on policy level perspectives was helpful in situating how individual level and system level factors are situated in this complex problem.

## Mental health promotion opportunities

Findings for mental health promotion opportunities from policy informants in this study differ from adolescent girl stakeholder perspectives from other studies which focus on leveraging aspects of spirituality, strengthening the ability of families and partners to act as a supportive unit; and implementing livelihoods opportunities to overcome food insecurity, purchase basic needs, access health services, and re-enroll in school [11, 32–35]. These findings indicate that policy makers focus more on direct modalities to delivery of mental health services while adolescent girls focus on aspects of their social ecology that can be strengthened to support their mental health. For policy makers, additional focus on modalities to strengthen supportive

relationships and address economic challenges may help to increase responsiveness to pregnant and parenting adolescent girls' preferences for mental health promotion opportunities.

## Recommendations for urgent policy

Given high rates of anxiety and depression, including suicidality among pregnant and parenting adolescent girls, urgent action to enhance policies that are more responsive to their needs is critical (Fig 5). WHO data suggests that the age-standardized suicide rate in Kenya was 5.6 per 100,000 population in 2016 [36]. Key informants broadly noted the lack of integration of mental health into ANC services. Policies that clarify mental health screening and support for all pregnant girls and women as a potentially lifesaving intervention with guidelines and resources for ANC providers to operationalize the policy is an important step. Building upon Kenya's existing policy framework, there are ample opportunities to strengthen the child and adolescent health policy, particularly in its implementation so that adolescent-friendly, free sexual and reproductive health and antenatal care services are implemented broadly across Kenya. There is some evidence that implementation of a Ministry of Health 28-item adolescent-friendly health services checklist in Kenya can lead to improved health outcomes for HIV positive adolescents, and that training providers in its use increases availability of adolescent service provision [37]. A similar approach to the Ministry of Health's for pregnant and parenting adolescent girls can be considered. Additionally, key barriers to full implementation of the adolescent friendly health service policy should be fully assessed and addressed including potential adaptations for ANC settings.

Kenya's current mental health policy (2015–2030) takes a life course approach to respond to the mental health needs of Kenyans [38]. The needs of pregnant and parenting adolescent girls are not specifically included within the mental health policy. Mental health prevention and promotion opportunities identified by key informants, including policy makers in this study, and pregnant and parenting adolescent girls should be considered for integration into

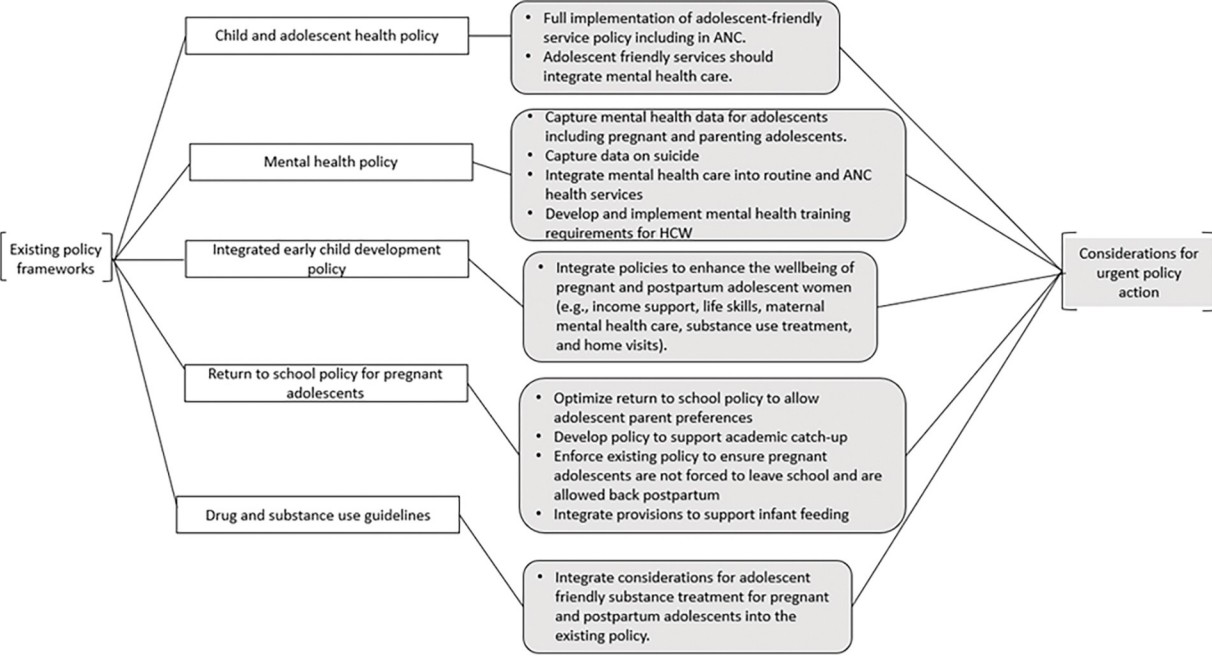

**Fig 5. Recommendations for urgent policy action.**

the existing mental health policy. Such opportunities include considerations for strengthening reporting on mental health indicators including suicidal ideation and attempts for adolescents including those who are pregnant and parenting. Policies that support mental health integration into adolescent-friendly services including ANC and establish mental health training requirements for providers may also be beneficial. As identified by policy informants, support for implementation of the policies is critical.

A 2018 UNICEF study examining Kenya's 2006 early childhood development policy identified opportunities to further strengthen the wellbeing of pregnant women and infants through strengthening the health system that serves them as well as their social ecosystem including families, partners and communities [39]. Suggestions to enhance the wellbeing of pregnant women included income support, life skills training, maternal mental health care, treatment for harmful substance use, and home visits which were thought to have potential benefits for the children as well. While the study did not focus on pregnant and parenting adolescent girls, the findings align with the perspectives of policymaker informants from this study and with adolescents from our other study arm and other studies in Kenya [12].

Our adolescent study findings indicate that inability to re-enroll in school which is most often due to economic challenges and required childcare responsibilities can contribute to feelings of depression. Conversely, a recent study in Kenya also found that the presence of depression reduces school attendance [40], underscoring the importance of mental health service provision for pregnant and parenting adolescent girls as a potential boost to increase education access and life opportunities including employment and socioeconomic status. Policy informants noted that a postpartum school re-enrollment policy is in place. However, discussions with adolescent girls, and other studies indicate that the policy requires strengthening to ensure that pregnant adolescents are not discharged from school due to their pregnancy, and that they are welcomed back in school postpartum. A recent study in South Africa found that a desire for earlier return to school than national policy currently allows was common [41]. Examination of Kenya's return to school policy may consider provisions to optimize timing for return to school based upon the mother's preferences. Policy measures that support adolescent mothers to catch up with their classmates should also be considered.

Both policymakers and adolescent girls indicated that breastfeeding is a challenge for returning adolescents. A thorough examination of the infant feeding policy in the context of maternal return to school may identify measures to integrate flexibility for adolescent mothers while optimizing their return to school including adapting feeding guidelines for adolescent mothers, provision of breast pump equipment, and space for feeding and/or pumping [41].

The government of Kenya released National Guidelines on Drug Use Prevention in 2021. The guidelines highlight age-appropriate prevention policies and highlight standards for inclusivity and non-discrimination, and identify interventions for family, school, workplace and community programs [42]. The 2017 National Protocol for Treatment of Substance Use Disorders in Kenya provides detailed information on the unique treatment needs of pregnant women [43]. Further information on adolescent friendly substance use treatment for pregnant and parenting adolescent girls may further enrich the policy.

## Limitations

While this study is comprehensive in nature, it is limited to the perspectives of 13 policymakers. Among the key informants in this study, we reached saturation point to the extent that we were receiving similar information across interviewees. However, it is possible that different perspectives may have been introduced were more key informants included. The study covers policy appraisal through a qualitative inquiry and perhaps other methodological

approaches such as consensus building or priority setting exercises can help sequence policy needs and action around this theme better. We may have also missed out on other policy makers and administrators covering other Ministries and dockets that may have bearing on adolescent health or mental health. Our respondents were receptive to the issues around perinatal and adolescent mental health and everyone we approached agreed to be interviewed. There may be a variability if a wider pool of policy advocates are considered. However, this pool of respondents were a highly receptive group of policy-engaged and responsive to mental health. This study revealed a critical gap in theoretical frameworks that specifically focus on mental health in low-income country settings. While we tried to address this through developing an integrated theoretical framework, the framework has not been tested. Future research should explore developing and testing a theoretical framework that can be applied in low-income country settings that is relevant to mental health policy.

## Conclusions

To support health outcomes for pregnant and parenting adolescent girls and their infants, a thorough examination of existing policies that influence their life experiences during this critical life stage is required to identify adaptations that can support their mental health. Such policies should increase the availability of adolescent-friendly services that integrate mental health into ANC and primary care, address economic challenges, strengthen the social ecosystem of pregnant and parenting adolescent girls, and support school re-enrollment. Further support to implement policies that seek to enhance adolescent mental health is also needed. Our policy informants unilaterally agreed that there was a lot to do around careful implementation of adolescent health policies to 'leave no one behind'.

## Acknowledgments

Authors would like to thank other mentors in INSPIRE Kenya's work, and policy makers who participated.

## Author Contributions

**Conceptualization:** Vincent Nyongesa, Marcy Levy, Joanna Lai, Manasi Kumar.

**Data curation:** Georgina Obonyo, Vincent Nyongesa, Joseph Kathono, Shillah Mwaniga, Manasi Kumar.

**Formal analysis:** Vincent Nyongesa, Joseph Kathono, Darius Nyamai, Manasi Kumar.

**Funding acquisition:** Marcy Levy, Joanna Lai.

**Investigation:** Shillah Mwaniga, Joanna Lai, Manasi Kumar.

**Methodology:** Malia Duffy, Shillah Mwaniga, Obadia Yator, Marcy Levy, Joanna Lai, Manasi Kumar.

**Project administration:** Georgina Obonyo, Vincent Nyongesa, Shillah Mwaniga, Obadia Yator, Marcy Levy, Joanna Lai, Manasi Kumar.

**Resources:** Joanna Lai, Manasi Kumar.

**Software:** Darius Nyamai.

**Supervision:** Joseph Kathono, Shillah Mwaniga, Obadia Yator, Manasi Kumar.

**Validation:** Joseph Kathono, Darius Nyamai, Obadia Yator.

**Visualization:** Georgina Obonyo, Vincent Nyongesa, Malia Duffy, Joseph Kathono, Obadia Yator.

**Writing – original draft:** Malia Duffy, Manasi Kumar.

**Writing – review & editing:** Vincent Nyongesa, Shillah Mwaniga, Marcy Levy, Joanna Lai, Manasi Kumar.

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
