## [Decision Letter · Decision Letter 0]

19 Oct 2022

PGPH-D-22-00958

Diverse policy maker perspectives on the mental health of pregnant and parenting adolescent women in Kenya: Considerations for comprehensive, adolescent-centered policies and programs

Dear Dr. Kumar,

Thank you for submitting your manuscript to PLOS Global Public Health. After careful consideration, we feel that it has merit but does not fully meet PLOS Global Public Health’s publication criteria as it currently stands. Therefore, we invite you to submit a revised version of the manuscript that addresses the points raised during the review process.

Your manuscript has been assessed by two expert reviewers, whose comments are appended below. The reviewers have highlighted concerns about several aspects of the manuscript including the reporting of the methodology, presentation of the results and the interpretation of the findings. Please ensure you respond to each point carefully in your response to reviewers document, and modify your manuscript accordingly.

We look forward to receiving your revised manuscript.

Kind regards,

Joseph Donlan

Editorial Office

Journal Requirements:

1. Please send a completed 'Competing Interests' statement, including any COIs declared by your co-authors. If you have no competing interests to declare, please state "The authors have declared that no competing interests exist". Otherwise please declare all competing interests beginning with the statement "I have read the journal's policy and the authors of this manuscript have the following competing interests:

a. Please clarify all sources of funding (financial or material support) for your study. List the grants (with grant number) or organizations (with url) that supported your study, including funding received from your institution. 

b. State the initials, alongside each funding source, of each author to receive each grant.

3. Your current Financial Disclosure states, “the funder has played no role”. However, your funding information on the submission form indicates that you received funding from “Fogarty International Center”/that you did not receive funding. Please indicate by return email the full and correct funding information for your study and confirm the order in which funding contributions should appear. Please be sure to indicate whether the funders played any role in the study design, data collection and analysis, decision to publish, or preparation of the manuscript.

4. Please provide separate figure files in .tif or .eps format only and remove any figures embedded in your manuscript file. Please also ensure that all files are under our size limit of 10MB.

5. In the online submission form, you indicated that "this is qualitative data with sensitive information but upon reasonable inquiry from the corresponding author can be made available upon request". All PLOS journals now require all data underlying the findings described in their manuscript to be freely available to other researchers, either 1. In a public repository, 2. Within the manuscript itself, or 3. Uploaded as supplementary information.

Additional Editor Comments (if provided):

Reviewers' comments:

Reviewer's Responses to Questions

**Comments to the Author**

1. Does this manuscript meet PLOS Global Public Health’s publication criteria? Is the manuscript technically sound, and do the data support the conclusions? The manuscript must describe methodologically and ethically rigorous research with conclusions that are appropriately drawn based on the data presented.

Reviewer #1: Yes

Reviewer #2: Yes

2. Has the statistical analysis been performed appropriately and rigorously?

Reviewer #1: N/A

Reviewer #2: Yes

3. Have the authors made all data underlying the findings in their manuscript fully available (please refer to the Data Availability Statement at the start of the manuscript PDF file)?

Reviewer #1: Yes

Reviewer #2: No

4. Is the manuscript presented in an intelligible fashion and written in standard English?

Reviewer #1: Yes

Reviewer #2: Yes

5. Review Comments to the Author

Reviewer #1: This manuscript examines an interesting topic, namely the focus on improving the mental health of pregnant adolescent girls and mothers through policy change and mobilization of stakeholders on all levels.

However, there are a number of challenges in the manuscript in this current form that the authors should consider reviewing.

Author Affiliations:

• The same subscript should be used for authors that are affiliated with the same institution.

Abstract

• In the first line, it should say, “adolescent girls” rather than “adolescents”.

• I would suggest using the term “girls” rather than “women” throughout the paper as the term women is commonly used among adults rather than adolescents.

• Make sure you are consistent in your use of adolescent women (girls) as the use of adolescent alone can make readers assume boys are also included.

Introduction

• Page 3, line 52 and line 55: Drop the “clear” as it’s not needed. I might suggest saying “Given the evidence…”

• Page 3, line 62: Perhaps say “East African” rather than “East Africa”

• Page 3, line 62: what is the “stigma and shame” related to? Falling pregnant at an early age or engaging in sexual activity before marriage? Please make it clearer what you mean by this statement and consider drawing upon prior literature.

• Page 3, line 67: You use the term “adolescent girls” however, previously you were using “adolescent women”. Again, I would suggest consistence and would use the term “adolescent girls”.

• Page 4, line 71: I suggest using the terms, “pregnancy and postpartum” periods.

• Page 4, line 73: drop the “we” and suggest saying “the inequities encountered in this population”.

• Page 4, line 81, remove the “-“ . The sentence is difficult to follow – I suggest revising.

Methods

• Page 5, line 113: Should say on “a password…”

• Page 5, line 114-115: Before you reached saturation, how many participants were recruited to take part in the study?

• Page 6, line 125: What is “VN”?

• Page 8, line 150. I suggest saying “discussed the interviews…”

• Can you provide more information on which language the interviews were conducted in along with how questions were formulated? Was there a semi-structured guide for the interviewers to follow?

• How were written consents obtained?

• In the data collection section, you use a lot of acronyms, however, it’s unclear on what you are referencing. If using the acronym for the first time, please spell it out first and put the acronym in parathesis so the readers understand what is being referenced when they encounter the acronym later in the paper.

• Overall, I would suggest taking a closer look at your methods as there seems to be some redundancy in the various sections.

Theoretical Framework

• This overarching theoretical framework is helpful for the reader and helps distill the multiple organizing principles (CFIR, etc) into a more streamlined approach

• Page 9, line 158: a different citation style is used – ensure consistence throughout the paper.

Results

• In terms of the presentation of quotes, it would be preferable to have an introductory sentence for each quote and, as helpful, a conclusion sentence. The current presentation of multiple quotes in a row does not help guide the reader enough – it will be important to point out what you’d like them to notice in each quote by paraphrasing. If the quotes discuss the exact same point, kindly chose one illustrative quote rather than presenting 2-3 on the same topic.

• For the quotes, could you kindly distinguish who said what by adding the gender of the participant, age (if available), etc. I suggest classifying the participants for example, “P1” for participant want one, “P2” , P3….P13. This would be helpful for the readers to understand that the supporting quotes are not from one individual. Also, there seems to be punctuation marks missing for most quotes.

• Page 11, line 212: Should say “adolescent girls face challenges accessing information…”?

• On page 12, please provide more specificity around the types of stigma experienced. As above, is this about taboos around young people having a sex life, the shame of actually falling pregnant, or something else?

• What do respondents mean when they say “the health workers there will not treat them well”? Is this about ignoring, verbal abuse, physical abuse, or some other kind of treatment?

• The quote in line 260 and 261 seems really central to the overall paper. Can you spend a bit more time on this with some reflective or critical commentary on what this means for maternal health overall, and how it places adolescent pregnant patients at considerable risk?

• Page 14, line 271: Can you say more about “commodities” that are not covered in the ANC free services.

• Can you critically reflect on the desire for youth friendly services, given the gap in skills, adequate training, and ongoing supervision for health workers? It’s one thing to say “we need youth friendly services” but quite another to actually empower health workers to do this at high quality in ways that overcome existing high levels of stigma.

• I don’t believe Table 2 is necessary for the paper, though you could consider for online supplemental appendix.

Discussion

• Page 25, line 511. Please provide the reference where the publication of the study with adolescent girls perspectives was published.

• Page 25, line 519: You say young parents which encompasses adolescent fathers – however, I believe you are only referring to adolescent mothers? If so, please make sure there are distinctions when discussing adolescent mothers, fathers, or both.

• Page 26, line 533: The sentence that starts with “school attendance was only brought up…” needs clarification.

• Page 29, line 602: “welcomed” rather than “welcome”.

• As so many of the thematic findings centered on stigma, it would be appropriate to craft a paragraph on how these findings speak to other stigma literature (HIV has a strong set of work, as does mental health and sexual reproductive health).

• Authors discussed the stigma related to adolescent pregnancy among family and overall community, however, I wonder whether there is stigma within communities regarding mental health and whether that could impede this population’s ability to seek services even if accessible?

• Recommendation for current policy section seems a bit long for peer-reviewed literature. I might suggest 2 paragraphs maximum and consider omitting the figure.

• Limitations: I also wonder whether the authors could discuss the challenges that might be encountered implementing this policy or the return to school policy in rural settings versus urban settings.

• Limitations: May you kindly explicitly note the limitations of your sampling approach in terms of how sensitized respondents were to the adolescent girl and pregnancy issues. For example, it seems plausible that respondents with no awareness of this issue would have declined to be interviewed. This is not a problem, but may suggest that the policy / programming suggestions are reflective of a more highly-engaged group of actors and work to bring other actors to awareness of these issues will be important.

Reviewer #2: Diverse policy maker perspectives on the mental 1 health of pregnant and parenting adolescent women in Kenya: Considerations for comprehensive, adolescent-centered policies and programs

Thank you for you for the opportunity to review this manuscript. The authors should be commended for exploring such an important topic. This is an impressive manuscript. I have attached some minor comments below.

Introduction

It would be good to quantify rates of poor mental health among pregnant adolescents e.g. see Roberts et al 2021.

Please reference the 2007 Reproductive Health Policy

Please define the age range for adolescents in the study.

Methods

Please provide a breakdown of how many health, social, educational etc. stakeholders are included in the study and some descriptive information of informants if possible (e.g. how long have they worked in the role, are they client facing etc?)

Please provide a refusal percentage.

Please provide further information on how conflicts between the research team were resolved.

Results

You have identified suicidality and child development issues as a particular issues for adolescents and their children in this population. Do you have any information about referral pathways utilised in these situations?

6. PLOS authors have the option to publish the peer review history of their article (what does this mean?). If published, this will include your full peer review and any attached files.

**Do you want your identity to be public for this peer review?** For information about this choice, including consent withdrawal, please see our Privacy Policy.

Reviewer #1: No

Reviewer #2: No

---

## [Decision Letter · Decision Letter 1]

5 May 2023

Diverse policy maker perspectives on the mental health of pregnant and parenting adolescent women in Kenya: Considerations for comprehensive, adolescent-centered policies and programs

PGPH-D-22-00958R1

Dear Dr. Kumar,

We are pleased to inform you that your manuscript 'Diverse policy maker perspectives on the mental health of pregnant and parenting adolescent women in Kenya: Considerations for comprehensive, adolescent-centered policies and programs' has been provisionally accepted for publication in PLOS Global Public Health.

Best regards,

Ahmed Waqas

Academic Editor

Reviewer Comments (if any, and for reference):

Reviewer's Responses to Questions

**Comments to the Author**

1. If the authors have adequately addressed your comments raised in a previous round of review and you feel that this manuscript is now acceptable for publication, you may indicate that here to bypass the “Comments to the Author” section, enter your conflict of interest statement in the “Confidential to Editor” section, and submit your "Accept" recommendation.

Reviewer #3: All comments have been addressed

2. Does this manuscript meet PLOS Global Public Health’s publication criteria? Is the manuscript technically sound, and do the data support the conclusions? The manuscript must describe methodologically and ethically rigorous research with conclusions that are appropriately drawn based on the data presented.

Reviewer #3: (No Response)

3. Has the statistical analysis been performed appropriately and rigorously?

Reviewer #3: N/A

4. Have the authors made all data underlying the findings in their manuscript fully available (please refer to the Data Availability Statement at the start of the manuscript PDF file)?

Reviewer #3: Yes

5. Is the manuscript presented in an intelligible fashion and written in standard English?

Reviewer #3: Yes

6. Review Comments to the Author

Reviewer #3: Thank you for providing me with the opportunity to review this manuscript. I have reviewed the manuscript and associated revisions based on previous reviewers' comments.

The paper addresses an important and timely issue by exploring the mental health of pregnant and parenting adolescent women in Kenya. This research contributes to the growing body of literature on the mental health of vulnerable populations and will be valuable in informing future policy and programmatic efforts. The paper is methodologically sound, employing rigorous qualitative research methods to gather and analyze data. The authors' clear explanation of their research design ensures the credibility and trustworthiness of the study's findings.

The paper effectively highlights the practical implications of the findings for policy makers and program developers. By offering concrete recommendations for comprehensive, adolescent-centered policies and programs, the authors ensure that their research has a direct and positive impact on the lives of pregnant and parenting adolescent women in Kenya.

7. PLOS authors have the option to publish the peer review history of their article (what does this mean?). If published, this will include your full peer review and any attached files.

**Do you want your identity to be public for this peer review?** For information about this choice, including consent withdrawal, please see our Privacy Policy.

Reviewer #3: **Yes: **Dr Hafsa Meraj
